# An automated methodology for differentiating rock from snow, clouds and sea in Antarctica from Landsat 8 imagery: A new rock outcrop map and area estimation for the entire Antarctic continent

Alex Burton-Johnson[1]*, Martin Black[1,2], Peter T. Fretwell[1] and Joseph Kaluza-Gilbert[3]

[1] British Antarctic Survey, Cambridge, CB3 0ET, UK
[2] Department of Geography, Environment and Earth Sciences, University of Hull, Hull, HU6 7RX, UK
[3] School of Geography, Earth and Environmental Sciences, University of Birmingham, Birmingham, B15 2TT, UK

*Correspondence to*: Alex Burton-Johnson (alerto@bas.ac.uk)

**Abstract.** As the accuracy and sensitivity of remote sensing satellites improve there is an increasing demand for more

accurate and updated base data sets for surveying and monitoring. However, differentiating rock outcrop from snow and ice is a particular problem in Antarctica where extensive cloud cover and widespread shaded regions lead to classification errors. The existing rock outcrop dataset has significant georeferencing issues as well as overestimation and generalisation of rock exposure areas. The most commonly used method for automated rock and snow differentiation, the Normalised Difference Snow Index (NDSI), has difficulty differentiating rock and snow in Antarctica due to misclassification of shaded pixels and

is not able to differentiate illuminated rock from clouds. This study presents a new method for identifying rock exposures using Landsat 8 data. This is the first automated methodology for snow and rock differentiation that excludes areas of snow (both illuminated and shaded), clouds and liquid water whilst identifying both sunlit and shaded rock, achieving higher and more consistent accuracies than alternative data and methods such as the NDSI. The new methodology has been applied to the whole Antarctic continent (north of 82°40' S) using Landsat 8 data to produce a new rock outcrop dataset for Antarctica.

The new data (merged with existing data where Landsat 8 tiles are unavailable; most extensively south of 82°40' S) reveals that exposed rock forms 0.18% (21,745 km$^2$) of the total land area of Antarctica; half of previous estimates.

## 1 Introduction

Differentiating areas of snow and exposed rock in Antarctica is important in a variety of contexts, including mapping, navigation, glaciological, geological and geomorphological research, and monitoring changes in the ice sheet and its

response to climate change. The only existing continent-wide geospatial dataset for exposed rock in Antarctica is available from the Scientific Committee on Antarctic Research (SCAR) Antarctic Digital Database (ADD) website, www.add.scar.org. This dataset (the ADD rock outcrop dataset, Thomson & Cooper 1993) has been derived through manual identification and digitization of published topographic maps. The dataset comes from a variety of sources of varying scales and accuracies, so the accuracy of the dataset is regionally inconsistent and has no quality assessment associated with it.

Although extensively used (over 2500 downloads of the rock outcrop dataset in the last 3 years, e.g. Riley et al. 2011, Golynsky et al. 2006, Vaughan et al. 1999) the data suffers from poor georeferencing, frequent misclassification of shaded snow as rock, as well as overestimating and generalising areas of exposed rock (Fig. 1). Additionally, as satellite derived coastlines and digital elevation models become available, the inconsistency and inaccuracy of the present cartographically derived ADD rock outcrop dataset becomes difficult to resolve with these new data sources. There is therefore an urgent need to improve the consistency georeferencing and accuracy of rock outcrop data for Antarctica.

In temperate regions methods have been formulated to automatically identify exposed rock outcrop from satellite imagery (e.g. Racoviteanu *et al.* 2010, Dozier 1989, Hall *et al.* 1995, Paul *et al.* 2002, Paul *et al.* 2009, Bolch *et al.* 2010, Zhu & Woodcock 2012, Zhu *et al.* 2015), but the methods have never been applied to Antarctica. The most commonly used existing method for delineating snow cover and rock outcrop is the Normalised Difference Snow Index (NDSI, Hall *et al.* 1995, Dozier 1989). The NDSI was developed following other indices, such as the Normalised Difference Vegetation Index (NDVI, Tucker 1986, Tucker 1979), initially for application to MODIS and Landsat satellite imagery. The NDSI is calculated according to equation (1) (modified for Landsat 8 data) where Landsat 8 OLI (the Landsat 8 Operational Land Imager sensor) band 3 equates to spectral wavelengths of 0.53 to 0.59 μm (the green band) and OLI band 6 equates to spectral wavelengths of 1.57 to 1.65 μm (the short wavelength infrared band, SWIR 1):

$$NDSI = \frac{OLI\ band\ 3 - OLI\ band\ 6}{OLI\ band\ 3 + OLI\ band\ 6} \tag{1}$$

Equation (1) works on the basis that snow reflects visible wavelengths stronger than middle-infrared wavelengths whilst rock displays a slightly higher reflectance for middle infrared wavelengths than visible wavelengths (Fig. 2) and so a threshold value can be determined for the NDSI of an image differentiating pixels of snow and rock (typically in the range 0.25 to 0.45, Hall *et al.,* 1995). One problem for application of the thresholded NDSI technique to automated snow and rock differentiation is that the optimal threshold value must be determined for each individual image being analysed or even varied within the same image due to changes in illumination or fresh snow cover across the image's area (Burns and Nolin, 2014). It is often the case that the optimal threshold is manually determined on each scene by comparison to reference data, however this becomes a problem when large numbers of images need to be analysed or reference data is not available.

Although the application of the NDSI has been successful at lower latitudes (e.g. Burns & Nolin 2014) where vertically illuminated imagery is available, high solar elevation angles in Antarctica lead to exclusion of shaded rock. This issue of shaded rock is greater in Antarctica where unavoidably low solar elevation angles result in large percentages of the outcrop being in shadow. The problem has been addressed for glacier mapping at lower latitudes by thresholding the Landsat blue band (in addition to an NDSI or alternative band ratio threshold) due to the higher reflectance of shaded snow than shaded rock in blue wavelengths (Arendt et al., 2012; Bishop et al., 2004; Paul et al., 2007; Paul and Kääb, 2005).

Unavoidable cloud cover in some Antarctic images, especially on the Antarctic Peninsula, leads to the classification of clouds as rock exposure by the NDSI technique (Fig. 3) as the two are indiscernible using this methodology. Any effective dataset of rock outcrop in Antarctica would have to ensure that clouds are not misrepresented.

A further problem for automated rock identification at lower latitudes is debris cover on glaciers which is indiscernible in multispectral imagery from exposed rock (Paul et al., 2004). This is accentuated by the melting and ablation of low latitude glaciers (Stokes et al., 2007) and is intensified by large amount of debris from frost shattering and freeze thaw activity (Fig. 4a and 4b). However, Antarctic glaciers are rarely debris covered due the prevailing climatic conditions where constant sub-freezing conditions result in a lack of ablation (Fig. 4c and 4d). The limited number of positive degree days and the lack of a day/night cycle at polar latitudes reduces freeze thaw activity meaning that less frost shattering takes place. Most Antarctic glaciers and ice streams are marine terminating and relatively few have active ablation zones (with the exception of a small percentage on the northern and eastern Antarctic Peninsula). The result is that most Antarctic glaciers are largely debris-free, removing this limitation from our study.

Here we present a new technique for automated rock outcrop identification using freely available Landsat 8 satellite data. The method is a composite technique combining separate algorithms that divide the image into cloud, liquid water, shaded snow and sunlit snow and shaded and sunlit rock exposures. We test the method against manually digitised polygons, the existing ADD rock outcrop dataset and the NDSI to validate and compare its accuracy.

We apply the new methodology to the entire landmass of Antarctica, (>12,000,000 $km^2$) using Landsat 8 data over all regions of the continent that contain rock outcrop. The resulting dataset represents an improvement over the previous dataset (ADD), providing consistent and accurate estimation of the amount and location of rock outcrop in Antarctica at 30 m resolution.

## 2. New Methodology

### 2.1 Input data

To produce a rock outcrop map for the entire Antarctic continent requires a freely available georeferenced multiband dataset. The dataset must cover high latitudes; be recently acquired; be of a high enough resolution to identify individual outcrops and geomorphological features; and be divided into sufficiently large scenes to allow for manual selection of suitable tiles for the entire continent. On this basis, the Landsat 8 multispectral satellite data was chosen for analysis. Landsat 8 is the latest and continuing satellite mission for multispectral global data acquisition launched by NASA and the United States Geological Survey (Roy et al., 2014). The satellite's Operational Land Imager (OLI) sensor records 8 electromagnetic bands (0.43 to 2.29 μm wavelengths) at 30m resolution, plus a panchromatic band (0.50 to 0.68 μm) at 15m resolution, whilst its Thermal Infrared Sensor (TIRS) records two thermal infrared bands (TIRS1 and TIRS2, 10.60 to 12.51 μm) acquired at 100m resolution and resampled to 30m. However, the TIRS has suffered from calibration issues and whilst calibration

changes have been made to some of the TIRS1 datasets, the TIRS2 data has a larger and more variable calibration uncertainty. Consequently only TIRS1 data is used in this study.

For the production of an Antarctic-wide rock outcrop map, tiles were selected that display strong illumination and minimal cloud cover. To ensure strong illumination we only used images taken during the day in the Austral summer between

September and March, with all but 17 images having solar elevation angles >20°. An estimate of cloud cover is included in the metadata for Landsat 8 images, with <30% cloud cover for all but four of the images used. Of particular importance when selecting suitable images was to exclude tiles with extensive cumulus or stratocumulus cloud where shadows within and below the cloud layer can be indiscernible from illuminated rock exposure. A total of 249 Landsat 8 tiles meeting these requirements were identified using the USGS Earth Explorer website (earthexplorer.usgs.gov). The images used were

acquired between October 2013 and March 2015 (details of the images used are included in the supplementary data). Most areas are covered by multiple tiles, increasing the procedure's sensitivity, reducing the effect of variable snow cover and allowing outcrops to be found in areas masked by cloud in one of the composite images. Details of the tiles used are provided as supplementary material.

In addition to the raw data, pre-processed tiles (170 km North-South by 183 km East-West) corrected for top of atmosphere

reflectance, surface reflectance and brightness temperature are freely available for download (espa.cr.usgs.gov). However, the calculation of surface reflectance values in Antarctica is problematic due to a lack of adequate atmospheric correction models for the continent, limited *in situ* atmospheric data and inadequate quality elevation data (Black et al., 2014), rendering surface reflectance corrected data unsuitable. Instead, top of atmosphere reflectance corrected and brightness temperature converted products were used, as were also used for the Landsat Image Mosaic of Antarctica (Bindschadler et

al., 2008).

## 2.2 Methodology

The new methodology identifies areas of sunlit and shaded rock through two separate workflows and then merges both outputs to produce the final dataset. Within both procedures a series of masks are produced to identify areas of exposed outcrop and to exclude areas of snow, cloud and liquid water. At each stage band ratios were used in preference to threshold

values for individual bands to allow application of a single set of threshold values to a large dataset. These two procedures are detailed below and a flowchart for executing this process shown in Fig. 5. The complete methodology was automated within ArcPy (Zandbergen, 2013). The script is available from GitHub (github.com/mblack2xl/AntarcticRockOutcrop). Note that raster calculations use raster values already corrected for surface reflectance and converted to brightness temperature as downloaded from the ESPA website. Raster values in these Landsat products are scaled for storage as 16-bit integers using

the following scale factors: 0.0001 for TOA bands 1 to 9, and 0.1 for bands 10 and 11 (e.g. a band 2 blue reflectance of 0.25 will be stored as 2500 and a band 10 TIRS1 brightness temperature of 255 K will be stored as 2550; Anon, 2016).

Threshold values used in the methodology were determined by manually classifying 8741 pixels from three different Landsat 8 images of different latitudes, geology, illumination and cloud cover. Pixels were classified as representing "clouds", "sea",

"sunlit rock", "shaded rock", "sunlit snow", or "shaded snow". Pixel values were extracted for the spectral bands of interest to determine the spectral properties of these six land cover classes (Fig. 6) with thresholds being set that best distinguished them.

**Procedure A. Sunlit Rock:**

**A.1. Sunlit rock identification: the NDSI**

Although the NDSI is unable to identify shaded rock and often misclassifies clouds as rock outcrop, it remains the best method for identifying regions of exposed sunlit rock. Consequently, it is the primary input for this methodology with a threshold value of <0.75 being used to identify pixels of sunlit rock outcrop and confidently exclude pixels of snow (the upper 95[th] percentile of the range of values for sunlit rock, Fig. 6a).

**A.2. Cloud mask: TIRS1 / Blue and TIRS1 Threshold**

One of the main problems of rock outcrop identification in Antarctica is that sunlit rock and clouds are indiscernible using the NDSI alone (Fig. 6a). Consequently we have derived a mask for sunlit snow and clouds using the thermal infrared band (Landsat 8 TIRS1, 10.60 to 11.19 μm) and the blue band. Using a ratio of these bands, clouds and sunlit snow give low values as they are cold but have high blue reflectance (Fig. 6b). In contrast, pixels of sunlit and shaded rock are warmer

when associated with high blue reflectance or colder when associated with low blue reflectance, resulting in high to moderate ratio values. However, shaded snow and liquid water also give high to moderate values. Using the scaled Landsat 8 images a TIRS1/blue threshold value of >0.4 (>400 for non-scaled TIRS1 brightness temperature and blue reflectance values) is most effective in selecting cloud free pixels and excluding pixels of sunlit snow and cloud to produce an accurate final product, although some sunlit rock pixels are also discarded (Fig. 6b). This threshold represents the upper 95[th]

percentile for cloud and sunlit snow pixels. To aid this cloud masking further an absolute TIRS1 TOA brightness temperature threshold of >255 K (raster values of >2550 in scaled brightness temperature converted Landsat 8 images) is also applied as <1% of sunlit rock pixels have lower TIRS1 values whilst 10% of cloud pixels and 5% of sunlit and shaded snow pixels do have lower values (Fig. 6c).

**A.3. Liquid water mask: NDWI and coastline**

The most widely applied approach for the identification of liquid water in multispectral imagery is the Normalised Difference Water Index (NDWI, McFeeters 1996). Modified for Landsat 8 data with the Landsat 8 OLI band 3 equating to spectral wavelengths of 0.53 to 0.59 μm (the green band) and OLI band 5 equating to spectral wavelengths of 0.85 to 0.88 μm (the near infrared band, NIR) the NDWI is calculated using equation (2):

$$NDWI = \frac{OLI\ band\ 3 - OLI\ band\ 5}{OLI\ band\ 3 + OLI\ band\ 5}$$   (2)

A liquid water mask is applied to both the sunlit and shaded rock identification procedures to exclude liquid water offshore (seawater) and onshore (melt ponds), and so the same threshold value of <0.45 is used for both (Fig. 6d). Unfortunately, due to the presence of calved ice and suspended glacial debris in Antarctic coastal seawater, a large overlap in NDWI values

exists between pixels of sea and shaded rock exposure (Fig. 6d). To compromise between minimising the loss of shaded rock pixels whilst maximising seawater removal the NDWI threshold represents the upper 90[th] percentile for shaded rock pixels (Fig. 6d).  To aid this step the manually derived coastline of Antarctica (available from the SCAR Antarctic Digital Database, www.add.scar.org) was also used as a mask for excluding seawater and sea ice.

**Procedure B. Shaded Rock:**

**B.1. Shaded rock identification: Blue threshold**

Even in the shade, snow is more reflective at blue wavelengths than shaded rock. By comparing the blue reflectance values of pixels representing rock and snow a threshold reflectance value of <0.25 (raster values of <2500 in scaled TOA corrected Landsat 8 images) was found to successfully identify pixels containing shaded rock exposure. This threshold represents the intermediate value between the upper 95[th] percentile for shaded rock and the lower 95[th] percentile for shaded snow (Fig. 6e)

**B.2. Liquid water mask: NDWI and coastline**

Although a blue wavelength threshold successfully differentiates shaded snow and rock, liquid water is also misclassified as rock. Thus, the NDWI and coastline mask applied to the sunlit rock data are also applied to the shaded rock data (again using the NDWI threshold value of <0.45, Fig. 6d). This step also aids exclusion of shaded snow pixels as 25% of their values are discarded by the NDWI threshold (Fig. 6d).

**Procedure C. Applying the masks and merging the datasets**

Pixels that were identified as rock by the NDSI mask and not identified as cloud or water represent sunlit rock outcrops. Similarly, pixels with blue band intensities below the threshold for shaded rock that aren't subsequently identified as liquid water by the NDWI threshold represent shaded rock exposures. Merging these two outputs produced the rock outcrop map for each tile. Tiles not already projected with the WGS 1984 Stereographic South Pole spatial reference (i.e. those at scenes

with a centre latitude greater than or equal to -63° S, for example the South Shetland Islands) were then reprojected to this projection before the results of all the tiles were mosaicked together for the entire continent.

As most areas were covered by multiple overlapping Landsat tiles, any pixels identified as rock exposure by any of the overlying tiles was included as exposed rock in the final dataset. This was achieved by mosaicking the binary raster files produced by the workflow and taking the maximum pixel value. If a pixel was classified as snow it was designated "0" by

the script, or "1" if it represents rock. Consequently this mosaicking process stores rock outcrop pixels ("1") in the raster

mosaic in preference to snow ("0"). By analysing multiple overlapping tiles the methodology becomes more sensitive to identifying rock outcrops; allows detection of rock outcrops even when they are obscured by clouds in one tile of the input data; and makes the methodology less sensitive to seasonal or short term variation in snow cover.

Finally, the extent of the mosaicked raster dataset was converted in to a new polygon shapefile and merged with the existing ADD rock outcrop dataset for areas not covered by the Landsat 8 imagery (Fig. 7).

## 3. Results

### 3.1 Accuracy Assessment

To quantify the accuracy of the new methodology and its limitations the extent of rock exposure was manually delineated using ten 10x10 km images, totalling 1,000 km2 or 1,108,890 pixels (Fig. 8. Enlargements of the images in Fig. 8 can be downloaded from the supplementary material). Images were selected from distal locations across the continent (Fig. 9), covering a range in geology, geomorphology and latitude. Areas of rock outcrop were manually identified by three operators. One tile (Ryder Bay, Fig. 8g) was traced by all operators; operator variability for pixel identification (rock or non-rock) was ±0.27% (one standard deviation).

The manually derived land cover was compared with the existing ADD rock outcrop dataset, the new automated method and the optimum NDSI-determined output for each image. Optimum NDSI threshold values (maximum values for pixels identified as rock) were taken as those with the lowest total quantity disagreement (abundance accuracy) and allocation disagreement (location accuracy) (Pontius Jr. and Millones, 2011). As shown by Fig. 10, optimum NDSI threshold values are highly variable. For well illuminated images without any cloud cover (Fig. 8a to 8e), NDSI threshold values of 0.6 or 0.7 are optimal. Images of extensive shade achieve more accurate results at higher NDSI threshold values (0.8, Fig. 8f) allowing identification of shaded rock. In contrast, images with extensive thick cloud require lower values (0.3 to 0.5, Fig. 8g and 8h) so as not to include the cloud as misidentified rock outcrop pixels. Thinner, low clouds (Fig. 8i) are not so problematic and high values (0.7) remain optimal. For mixed images (Fig. 8j) with shaded and illuminated rock with minor cloud cover, 0.7 remained the optimal threshold value.

Well illuminated, cloud free images produce similar accuracies for the optimal NDSI technique and the new method (Fig. 8a to 8e) with low commission or omission disagreements (Fig. 11a). However, the required determination of an optimal NDSI threshold value renders this alternative methodology more involved than that used for our new dataset. In addition, even when using the optimal threshold value the NDSI technique omits areas of rock in shaded images as well as both shaded and sunlit rock in cloudy images, leading to high and variable omission disagreements (Fig. 11b).

The ADD rock outcrop dataset produces variable accuracies. In Ryder Bay (Fig. 8g) the map has been recently been updated using manual delineation from very high resolution aerial photography and so has high accuracy with low omission and commission disagreement, similar to the new dataset. However, it is important to stress that areas of high resolution outcrop mapping are limited in the ADD rock outcrop dataset. The ADD rock outcrop dataset is more accurate than the NDSI

technique in shaded images (Fig. 8f and 8j), but highly generalised and poorly georeferenced outcrop extents in other tiles (Fig. 8d, 8h and 8i) produce high and highly variable disagreements (Fig. 11), particularly in commission.

The new methodology performed poorest in images with limited areas of rock outcrop (e.g. Fig. 8h, 0.1% rock), although shade, clouds and mixed pixels of snow and rock in Fig. 8h make even manual pixel identification difficult. There are omission disagreements in shaded images (Fig. 8f and 8j) although these are much lower than for the alternative techniques (a mean of 15% for all images compared to 38% for the NDSI technique and 30 % for the ADD rock outcrop dataset, Fig. 11b). Clouds were successfully masked and do not contribute to the commission disagreement (Fig. 8h to 8j).

Mean statistics for the quality assessment are recorded in Table 1. The quality assessment shows higher accuracies for the new method (a mean of 85 ±8% 1SD error of correctly identified rock pixels for all ten images compared with 68 ±30% or 70 ±14% for the NDSI technique and ADD rock outcrop dataset respectively) with lower and much more consistent commission and omission disagreements than the alternative NDSI or ADD rock outcrop datasets (Fig. 11b). Using equation (3) to take commission and omission errors in to account, mean classification accuracies are 74 ±9% 1SD error for the new method compared to 63 ±27% for the optimal NDSI method or 39 ±19% for the existing ADD rock outcrop dataset (Fig. 11a).

$$Classification\ Accuracy = \frac{Correctly\ classified\ pixels}{Correctly\ classified\ pixels + Pixels\ of\ omission + Pixels\ of\ commission} \tag{3}$$

## 4. Discussion

This is the first automated methodology for the differentiation of snow and rock in Antarctica, from which a new outcrop map of the entire Antarctic continent has been produced at higher and more consistent accuracies than existing data and techniques (Fig. 11). The new dataset is available online via the SCAR Antarctic Digital Database (www.add.scar.org) and from this article's supplementary material.

Despite the poorer accuracy of the ADD rock outcrop dataset (39% classification accuracy compared to 74% for the new methodology, Fig. 11a), due to the methodology by which it was derived, certain features are better represented. This includes South Georgia and the South Orkney Islands where a lack of cloud-free imagery in the late Austral summer (when the outcrops aren't covered by snow) prevents automated outcrop identification. Consequently, rock outcrop extents in these areas are derived from the existing ADD dataset rather than remote sensing imagery, in addition to outcrops south of 82°40' S (Fig. 7).

It is important when using the new Landsat 8 rock outcrop map to consider seasonal variability in snow cover and that most outcrops were derived from multiple tiles from different years and different months of the Austral summer. As a result the map may not be representative of current conditions and may not consistently represent maximum outcrop extent across the continent.

## 4.1 Limitations

Using the new methodology we have produced a revised map of rock outcrops in Antarctica. Landsat 8 does not provide coverage south of 82°40' S so the existing ADD rock outcrop dataset was clipped to latitudes greater than this and merged with the new automatically derived data to produce the final dataset. There are two further limitations to the new

methodology:

1. Because an overlap exists between the NDWI values of shaded rock and liquid water (Fig. 6d) and because of inaccuracies in the existing coastal vector dataset, some pixels of coastal seawater not masked by the ADD coastline have been misidentified as exposed rock in all coastal scenes containing seawater pixels. This is particularly problematic for pixels adjacent to seawater rich in calved ice and glacial debris (Fig. 12a). These pixels are spectrally identical to shaded rock and

thus cannot be excluded automatically from the data. Consequently these pixels were manually removed from the final dataset, with the distinction of shaded rock and liquid water being made by eye. It should be noted that some of these misidentified pixels may still be present. However, as no manual editing was done on land the repeatability of this methodology should not be affected.

2. Even though spectral properties have been chosen that distinguish rock pixels from those of snow, clouds or sea, some

overlap exists where pixels remain ambiguous (Fig. 6). Consequently, to allow automated analysis over such a large area mildly conservative threshold values were chosen. For example, the NDSI threshold for sunlit rock was set at the 95[th] percentile rather than the complete range exhibited by sunlit outcrops as this excludes any overlap with the range of NDSI values for sunlit snow (Fig. 6a). This results in the exclusion of some pixels of exposed rock that are spectrally similar to clouds and snow (e.g. Fig. 12b).

3. Due to the 100m spatial resolution of the TIRS band, small outcrops around the continent (especially those less than 60m or 2 pixels across) are often excluded by the new technique and may be better represented in the ADD rock outcrop dataset.

4. Whilst Antarctic glaciers are rarely show any debris cover (Fig. 4) there are local occurrences where extensive debris cover does occur (most notably in the vicinity of the Dry Valleys and the NW coast of the Ross Ice Shelf) which are mapped as outcrop in the new dataset. However, it should be noted that these occurrences are isolated on the continental scale. As

this project aims to provide a consistent and automated approach that can be reproduced in the future (for example to monitor change in ice cover over time or season) our methodology attempts to be as free as possible from manual changes. We accept that in some areas localized occurrences of debris covered glaciers may need to be manually altered if detailed topographic maps of rock outcrop are required.

**4.2 Total outcrop area**

We calculate (using  the South Pole Lambert Azimuthal Equal Area projection) that the existing ADD rock outcrop dataset has a 44,900 km$^2$ area of rock outcrop, equivalent to 0.37% of the total land area of Antarctica (12,188,650 km$^2$, the SCAR

Antarctic Digital Database ). In contrast the new data has a 21,745 km$^2$ total area of rock outcrop (±5,654 km$^2$ based on the 74% calculated classification accuracy, section 3.1), equivalent to 0.18% ±0.05% of the continent's land area and 48% of the previous estimate. This is a significant decrease and highlights an overestimation in the current predictions of rock outcrop extent in Antarctica.

## 4.3 Applications and future developments

The new Landsat 8 rock outcrop map will provide a revised and accurate base dataset for future topographical, glaciological, geological and geomorphological mapping. A number of satellite programs collecting new high resolution colour images have recently been launched or are planned for launch in the near future, including the Digitalglobe WorldView-3 satellite (launched 2014), NASA's HyspIRI satellite (proposed but not yet under development), European Space Agency's Sentinel program (three satellites already launched with more under development) and the continuing Landsat data acquisition (continuing acquisition from Landsat 7 and 8, with Landsat 9 planned for launch in 2023). These new datasets will allow further application of this technique at higher resolutions and consequently higher accuracies, allowing future improvement of the datasets broader applications. Application of the new technique to these alternative datasets would however require modification of the threshold values for each mask in the procedure.

Once the available imagery has improved the Antarctic rock outcrop dataset will again be updated to exploit the new data and increase coverage of the continent (especially south of 82°40' S). By providing the code used in this study (available from GitHub, github.com/mblack2xl/AntarcticRockOutcrop) users will be able to apply the new methodology to their specific areas of interest and modify the thresholds for improved results on local scales. Such work may be possible to integrate in to future iterations of the continental dataset if users inform the authors regarding their new datasets.

## 5. Conclusions

A new map of exposed rock outcrop has been developed for the Antarctic continent. The new map was achieved via an automated methodology employing Landsat 8 multispectral imagery. The new methodology uses the NDSI technique to identify sunlit rock exposure and low blue intensities for shaded rock, and then applies separate masks to remove incorrectly classified pixels of cloud, snow and liquid water. This is the first automated methodology for rock outcrop identification in Antarctica, and achieves higher and more consistent accuracies than the existing dataset or what can be achieved using the alternative automated technique (the NDSI). Assessing the accuracy of these alternative techniques and datasets across a range of images gives a mean value for correct pixel identification of 85 ±8% for the new method compared to 70 ±14% using the existing ADD rock outcrop dataset or 68 ±30% for the NDSI technique using optimal values. Overall classification accuracies accounting for omission and commission errors improve from 39% for the existing ADD rock outcrop dataset and 63% for outcrop detection using optimal NDSI thresholds to 74% using the new technique.

The new map, supplemented by existing data for latitudes south of 82°40' S (the limit of Landsat 8 coverage), reveals that rock outcrop forms 0.18% ±0.05% (21,745 km$^2$) of the total land area of Antarctica, 48% of the previous estimate (0.37%, 44,900 km$^2$).

**Acknowledgements**

We would like to thank Allen Pope and our anonymous second reviewer for their positive, helpful and thorough reviews. This study is part of the British Antarctic Survey Polar Science for Planet Earth programme, funded by the Natural Environment Research Council (NERC). MB was funded by a NERC research studentship (NE/K50094X/1).

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

| Methodology | Correct % | SD | Commission % | SD | Omission % | SD | Classification Accuracy % | SD |
|---|---|---|---|---|---|---|---|---|
| This Study | 85 | 8 | 17 | 13 | 15 | 8 | 74 | 9 |
| Optimum NDSI | 68 | 30 | 7 | 6 | 32 | 30 | 63 | 27 |
| ADD Rock Outcrop | 70 | 14 | 154 | 212 | 30 | 14 | 39 | 19 |

**Table. 1. Summary of mean accuracy assessment vales for the ten images evaluated.**

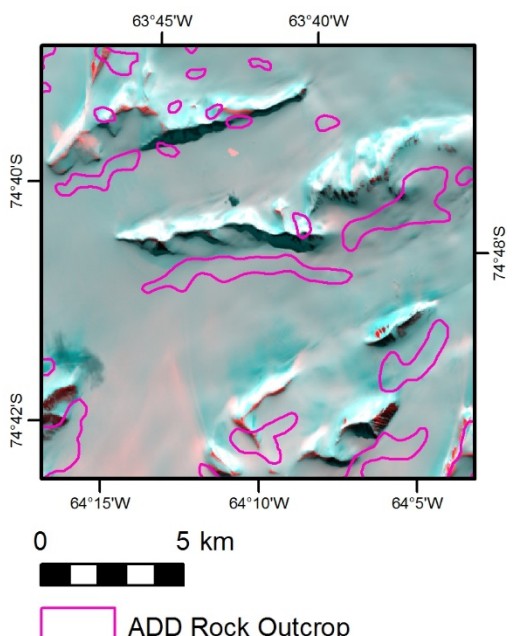

**Fig. 1. Example of the issues with the existing ADD rock outcrop dataset showing the problems with the georeferencing, overestimation and generalisation of areas of rock outcrop. The example uses a false colour image using the band combination Red: SWIR2; Green: Blue; and Blue: Blue. This combination accentuates rock, snow and cloud distinctions with red/pink pixels representing rock or clouds and turquoise pixels representing snow.**

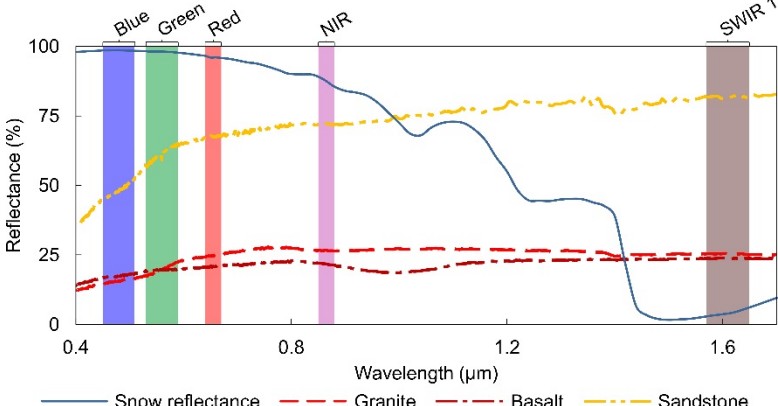

**Fig. 2. Spectral reflectance data for snow and rock (granite, basalt and sandstone) from the ASTER Spectral Library v1.2 (Baldridge et al., 2009). Designations of spectral regions as defined by the Landsat 8 bands: Blue – Band 2, 0.45 – 0.51 µm; Green – Band 3, 0.53 – 0.59 µm; Red – Band 4, 0.64 – 0.67 µm; NIR, Near Infrared – Band 5, 0.85 – 0.88 µm; SWIR 1, Short Wave Infrared – Band 6, 1.57 – 1.65 µm.**

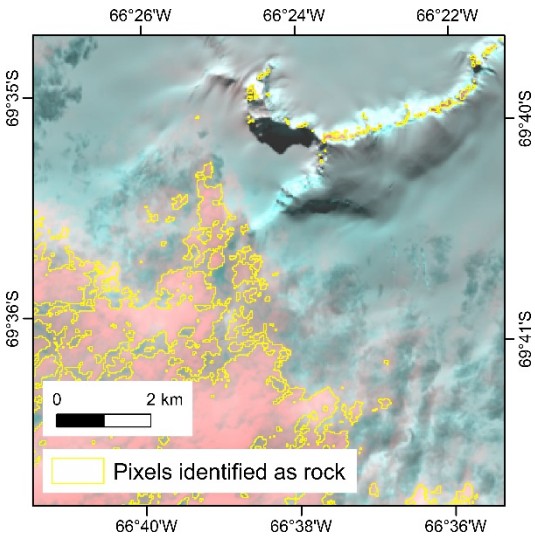

**Fig. 3. Illustration of the misclassification of cloud cover as rock pixels when using the NDSI technique. As in Fig. 1, the example uses a false colour image using the band combination Red: SWIR2; Green: Blue; and Blue: Blue. An NDSI threshold of 0.6 is used here to identify the rock outcrops, but at this threshold much of the cloud cover is also included. The Landsat 8 scene used is LC82161092014338LGN00.**

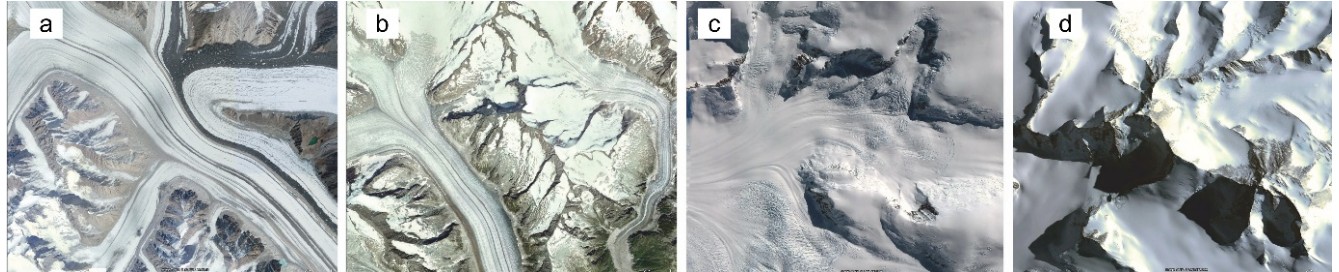

**Fig. 4. Comparison of debris cover for glaciers at low latitudes (3a, Karakoram Range (35°N), and 3b, Jungfrau Range, Alps (46°N)) with those of Antarctica (3c, Antarctic Peninsula (66°S), and 3d, Transantarctic Mountains (72°S)). Note the lack of surface moraine and the deep shadows in 3e and 3d, typical of Antarctic glaciers where a lack of day-night cycle and year-round low temperatures restricts freeze thaw action and the permanently low sun angles result in deep shadows in remotely sensed imagery.**

```
Download TOA corrected and brightness temperature
converted Landsat 8 tiles from espa.cr.usgs.gov and
the Antarctic coastline from www.add.scar.org
```

Identify sunlit rock ← → Identify shaded rock

Calculate the NDSI | Calculate TIRS1/Blue | | Calculate the NDWI | | Calculate the NDWI

Identify rock pixels by creating a new raster where NDSI <0.75

Identify cloud free pixels by creating a new raster where TIRS1/Blue >400

Identify pixels where TIRS1 >255 K to aid cloud masking

Identify liquid water free pixels by creating a new raster where NDWI <0.45 and pixels are classed as "land" by the ADD Antarctic coastline

Identify shaded rock by creating a new raster where Blue <0.25

Identify liquid water free pixels by creating a new raster where NDWI <0.45 and pixels are classed as "land" by the ADD Antarctic coastline

Add all masks together and create a new raster for sunlit rock that fulfils all four requirements

Add both masks together and create a new raster for shaded rock that fulfils both requirements

Merge the sunlit and shaded rock outcrop rasters → Ensure coordinate system is 'WGS 1984 Antarctic Polar Stereographic' → Repeat all previous steps for all Landsat images → Mosaic all Landsat rock outcrop rasters in to a new raster mosaic → Colvert raster mosaic to polygon

New Antarctic rock outcrop polygon ← Merge new outcrop polycon and clipped ADD outcrop polygon ← Clip ADD rock outcrop polygon to areas without Landsat coverage ← Manually remove remaining misclassified coastal seawater

**Fig. 5. Flowchart for the automated identification of rock outcrops in Antarctica. Threshold values are given without the 16-bit scaling used in the corrected Landsat 8 raster images.**

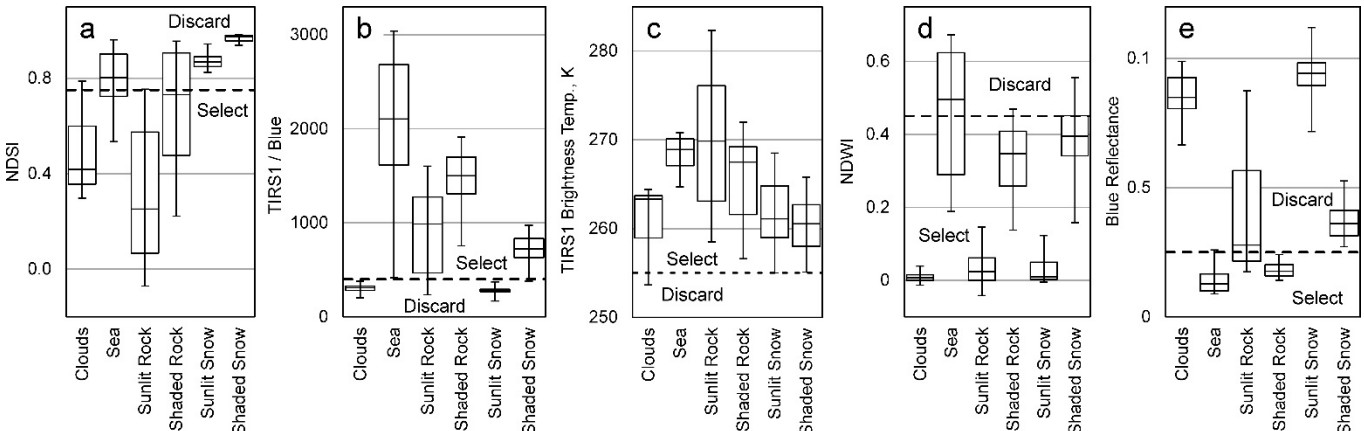

**Fig. 6. Box plots of extracted pixel values from three Landsat 8 tiles illustrating the different spectral properties of clouds (number of extracted pixels, *n* = 871), sea (*n* = 3277), sunlit rock (*n* = 1158), shaded rock (*n* = 1224), sunlit snow (*n* = 1293) and shaded snow (*n* = 918). TIRS1 brightness temperature, blue reflectance and the TIRS1/Blue values are converted from the scaled values of the**

5  **TOA corrected and brightness temperature converted Landsat 8 products (Section 2.2). Boxes indicate the 2<sup>nd</sup> and 3<sup>rd</sup> quartiles and median values. Whiskers indicate the 5<sup>th</sup> and 95<sup>th</sup> percentile. Dashed lines indicate the chosen threshold values for the automated rock outcrop extraction and the values to be selected or discarded.**

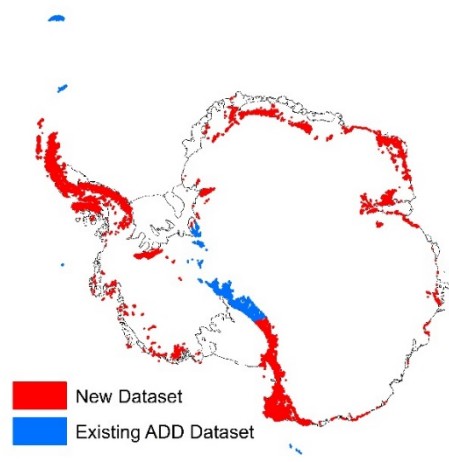

10  **Fig. 7. Rock exposure map of Antarctica showing the data sources for the new dataset. Outcrops shown in red were derived using the new remote sensing methodology and outcrops in blue were derived from the existing ADD rock outcrop dataset to supplement areas not covered by the Landsat 8 imagery (areas south of 82°40' S or islands lacking suitable cloud free images). Areas of rock exposure are exaggerated for illustration.**

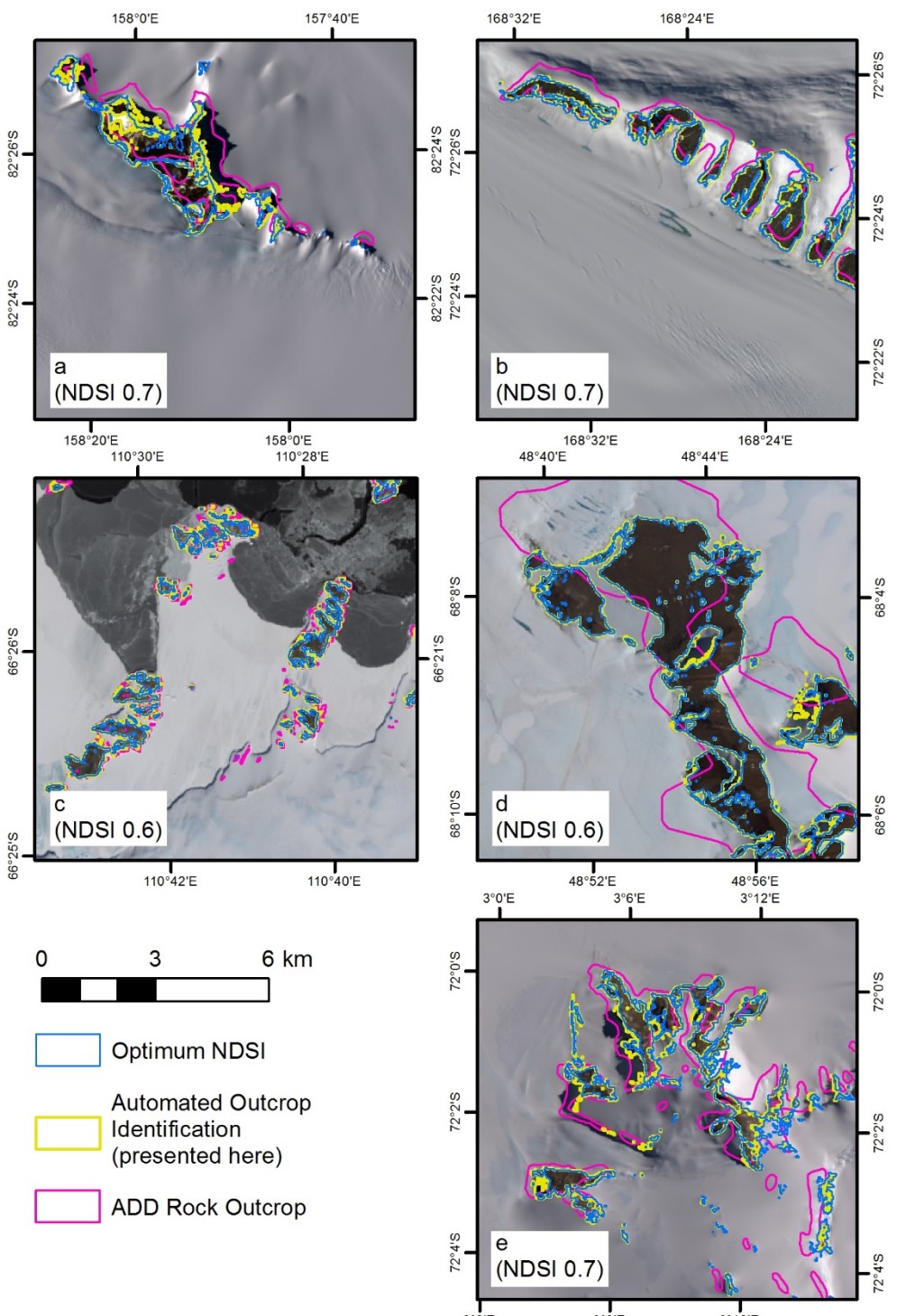

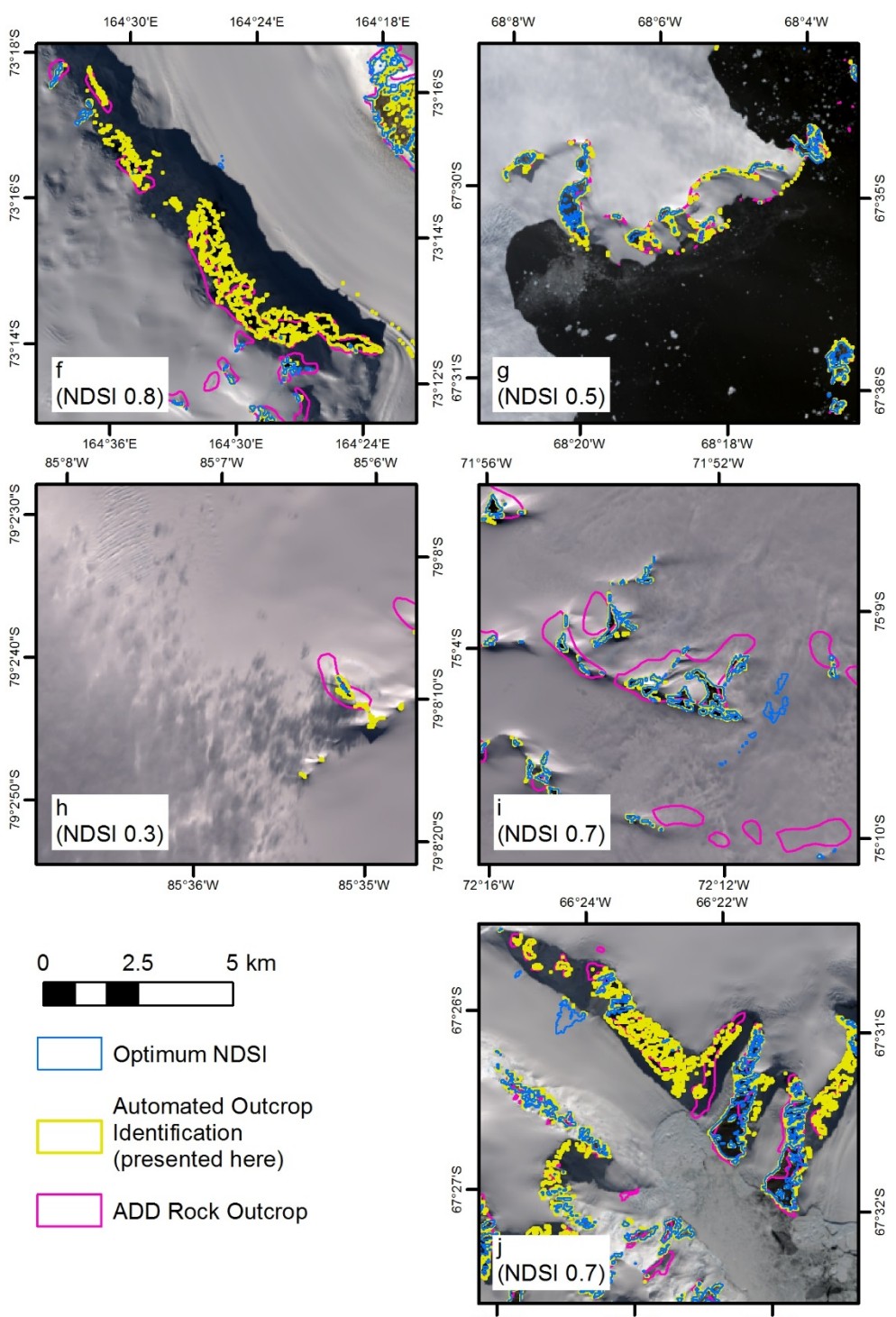

**Fig. 8.** Images used for the quality assessment overlain by the three alternative methodologies and datasets: Pixels extracted using optimum NDSI thresholds for each image (NDSI threshold values shown in brackets); pixels extracted using the new methodology presented here; and the extents of the current ADD rock outcrop map. Enlargements of these images can be downloaded from the supplementary material.

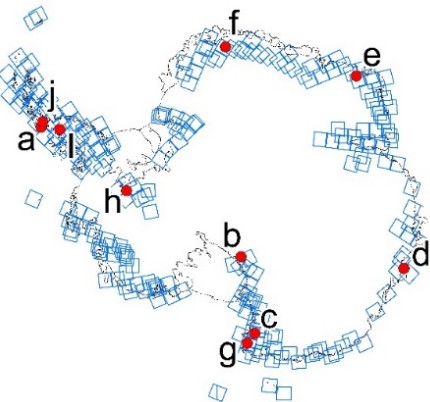

**Fig. 9.** Locations of the 249 Landsat 8 tiles (blue squares) used to identify rock outcrop in Antarctica and the locations (a to j) of the 10x10 km images used for the quality assessment in Fig. 8.

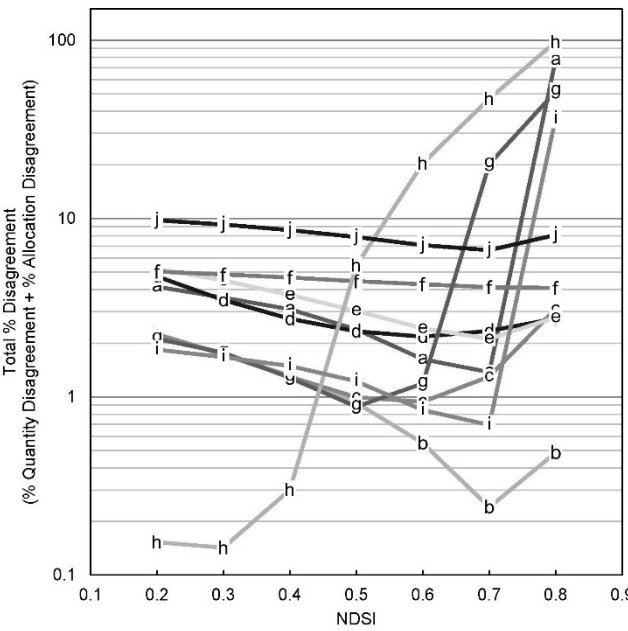

**Fig. 10.** Total quantity and allocation disagreement values (Pontius Jr. and Millones, 2011) for pixels extracted from the images in Fig. 8 using the NDSI threshold technique.

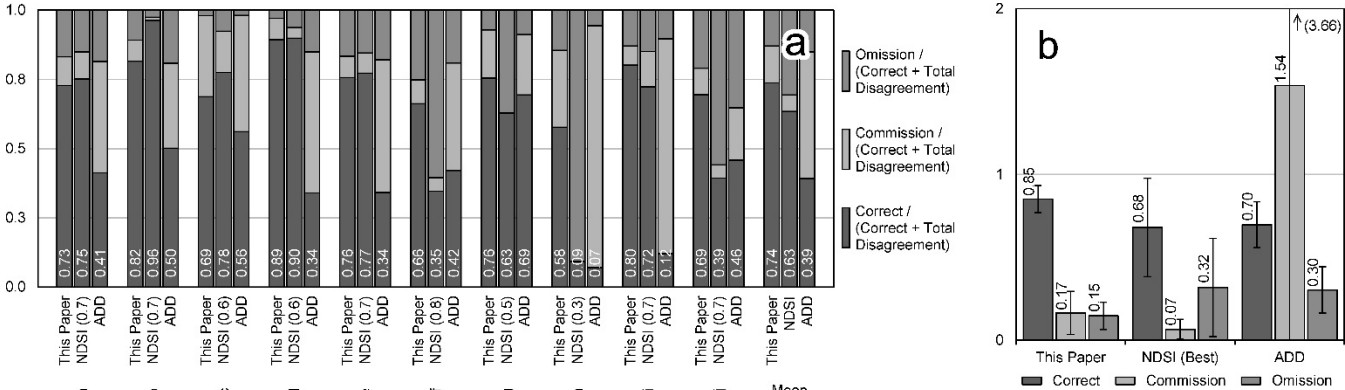

**Fig. 11. (a) 100% normalised accuracy assessment data for correctly classified pixels and pixels of omission and commission disagreements for the images in Fig. 8. Optimal NDSI values used are shown in brackets. Values in columns are the classification accuracy values: correctly classified pixels / (correctly classified pixels + total disagreement), equation (3) (b) Overall accuracy assessment data for the three alternative datasets showing the mean values for correct classification, commission and omission for the 10 QC images (Fig. 8) and error bars at 1SD.**

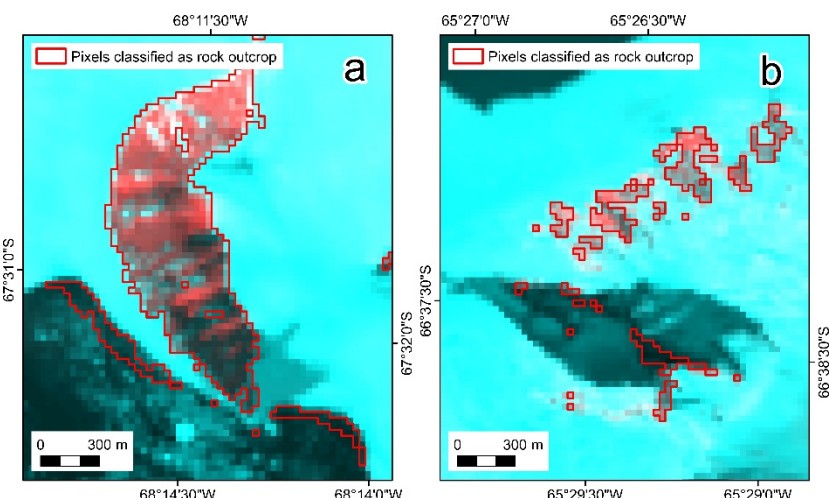

**Fig. 12. Examples of the new methodology's limitations, comparing the new dataset with false colour Landsat 8 images. The band combination (Red: SWIR2; Green: Blue; Blue: Blue) is chosen to accentuate rock, snow and seawater distinctions with red pixels representing rock, turquoise pixels for snow and dark green to black pixels for seawater. (a) An example of seawater near calving ice classified as rock (later removed manually). (b) An illustration of conservative outcrop extent estimation using the new technique. This is a result of the mildly conservative threshold values that had to be chosen to allow the automated analysis of so many tiles over the area of an entire continent.**