# Peer review of "An automated methodology for differentiating rock from snow, clouds and sea in Antarctica from Landsat 8 imagery: A new rock outcrop map and area estimation for the entire Antarctic continent"

_The Cryosphere, 2016_

## Referee Comment (RC1) · A. Pope (Referee) · 25 Mar 2016

General Comments

Burton-Johnson et al. present a new, largely automated method for identifying rock outcrops across all of Antarctica using Landsat 8 data. This paper contributes both a well-documented, open methodology as well as the resulting dataset. I wanted to especially commend the authors on their commitment to reproducible and open science.

[Figure]

The paper is clearly, concise, and open and will surely be widely used by the community (I know I will cite it when I update "Open Access Data in Polar and Cryospheric Remote Sensing", Remote Sensing, 2014). I look forward to seeing it fully published in The Cryosphere. Nevertheless, some edits could be made to make this method more applicable to future studies, to slightly expand the region over which Landsat data are applied, and to stylistically clarify a few points. See below for more specific comments.

Specific Comments

1) You include a short discussion of "how robust to a new sensor," but there are specific points you can include in the paper to better ensure this. Specifically, use units rather than DNs for thresholds (K, % reflectance), and also describe specifically how each threshold was set. For example, it appears that the NDSI threshold was the 95th percentile of sunlit rock. Including this would allow this method to be applied for future Landsats (to look for change), or with other sensors (to not rely on the ADD where Landsat coverage is unavailable).

2) The authors include a discussion and figure to illustrate the paucity of medial moraines in Antarctica. While this is largely true, I believe some manual cleaning must still be done. For example, in blue ice areas / the Dry Valleys, there may be some medial moraines which should be removed. A quick investigation of the Ferrar Glacier indicates this may be the case there.

3) Regarding extent of available Landsat 8 coverage – is there a mechanism in place to allow for updated data collections where Landsat 8 is now available rather than the ADD. Specifically: • A very quick EarthExplorer search showed that, for example, LC80021082016077LGN00 could possibly be used for Peter 1 Island. I would expect this to be the case for other sub-Antarctic islands. Please consider checking again for new Landsat 8 acquisitions, and discuss if there will be an updating procedure. • There have been very recent Landsat 8 off-nadir acquisitions which extend Landsat coverage further south (and more are planned for next year). Could you include these

in your analysis to reduce the areas where the ADD is needed? Will you be able to update next year to extend, too? Or is that prohibitive?

Technical Corrections

Title: This is not technically a fully automated method (manual cleaning is performed) – please edit title.

Title: "Landsat" should be "Landsat 8"

Abstract L9: I would suggest including an introductory sentence that provides motivation for the study.

Abstract L14: This is not a "fully automated" method – there is some manual cleaning. Please edit, perhaps to "largely" "nearly" or "overwhelmingly"

Abstract L18: May need to edit latitude (see Specific Comments)

Abstract L19: please indicate what physical areas 0.18% corresponds to.

P1 L21: I suggest citing appropriate examples where the ADD has been used.

P1 L26: Should "data" be treated as singular or plural?

P1 L27-28: Parenthetical note is perhaps presented too early for smooth flow of an introduction.

P1 L21-29: You should include some durable citation for the ADD if at all available. And/or a citation describing the dataset.

P2 L1: Consider adding a figure to illustrate the issues in the ADD dataset?

P2 L1: Comma after "Additionally"

P3 L8: "Landsat" should be "Landsat 8"

P3 L19: Follow parallelism in sentence structure – ". . .and be divided into sufficiently large scenes to allow for manual selection of suitable tiles for the entire continent."

P4 L1: Note that when building the Landsat Image Mosaic of Antarctica, Bindschadler et al. used TOA reflectance as well, given then largely dry and thin atmosphere over Antarctica. This could be a good supporting reference for TOA, if you wanted to include one.

P4 L10: I love that you're sharing your code! Perhaps also consider putting it on GitHub to allow sharing, editing, forking, etc.?

P5 Eqn2: You use the "TIRS" acronym, so to be consistent I think you should also use the "OLI" acronym, rather than referring to it as the Landsat 8 platform.

P5 L8: Please clarify whether "liquid water" refers to ocean, on land, or melt ponds on ice (or all of the above). It is a little confusing right now.

P5 L11-12: What is "blue intensity"? Reflectance? Some index?

P5 L23-24: Aren't all Antarctic satellite images already provided in Antarctic Polar Stereographic? Why is any reprojection necessaru?

P5 L26: "Landsat", not "LANDSAT"

P6 L3-6: Include total area used for this? What about an error matrix for this test?

P6 L11: reference formatting

P7 L14: See Specific Comments.

P7 L15: Have you considered including some quality information in the dataset, which indicates the date/image from which a particular polygon was digitized?

P7 L20: Somewhere in the discussion, it would be good to acknowledge the issues that TIRS has been having, its recalibration efforts, etc. This is important if your TIRS thresholds are at all sensitive...

P7 L27: What was used as "truth" for this manual removal?

P8 L7: Include space after "km2" and before "total"

P8 L10-17: See Specific Comments. The modification and selection of thresholds should be discussed more in depth in methods to facilitate this cross-application.

P8 L26: Period after "respectively" (not a comma)

P8 L27: May need to amend sentence with information about subantartic islands / off-nadir acquisitions (see Specific Comments).

P8 L27: Include are which 0.18% corresponds to.

P8 L28: "...48% of the previous estimate (0.37%, _____ km2)."

Figure 1: The snow reflectance line is solid in the legend but dashed in the figure. Make it solid in the figure. Consider using line colors from ColorBrewer to help colorblind readers.

Figure 2: Where did these data come from? Are the shadowed areas rock, it is hard to the viewer to see.

Figure 3: I don't find this figure very helpful – especially given Specific Comment regarding Ferrar Glacier. Could be removed or combined with other figure I suggest (illustrating issues with ADD).

Figure 4: In general, I don't see a systematic use of boxes and arrows to indicate particular inputs/outputs/actions. For example, rather than have an arrow pointing at "Identify Sunlit Rock" or "Identify Shaded Rock," these should be larger boxes (possibly shaded background) which encompass the specific tasks which achieve those goals. • In the top box "corrected" is not the right word, I don't think. "Pre-calculated" or just removed it, perhaps? • Relatedly, when are mask arrows being added or removed from each other? Clarify. • Units should be used either instead of or in addition to the DN values for TIRS and Blue • Remove superfluous steps (e.g. possibly reprojection, or "repeat steps for all Landsat images" • What method specifically was used for mosaicking Landsat rasters? What process used for overlapping areas? • What connectivity or filtering was used in polygonization? • Rather than clipping

[Figure]

ADD to a particular latitude, consider clipping to specifically areas of no Landsat? • Caption can deleted "using the new methodology" – this is implicit.

Figure 5: Regarding y axis units – some are clear (e.g. NDSI, NDWI), but it would be good to clarify the units/scaling factors for TIRS, Blue, etc. Also, again, what is "Blue Intensity"? Do you mean reflectance? Finally – how were test areas selected? And are "n" values pixels?

Figure 6: See Specific Comment regarding islands and coverage.

Figure 7: Include NDSI threshold either annotating figure (or in supplementary information if you think that would be too crowded). Also, consider organizing a-j by type of scene (or labeling as such) to make translation between text (P6 L11-16) and the structure of the figure easier. Two specific comments: "tree" should be "three" and "Automated Outcrop Identification" should be "Automated Outcrop Identification (presented here)"

Figure 8: How were areas chosen to not use any Landsat imagery? What if outcrops are showing up in new locations? Is there a method to handle that sort of change? Or inform selection of imagery from other sensors?

Figure 9: Should "Jr" have a period after it in the citation?

Figure 10: I would cut off the y axis of part b at 2 or a little below and include a label to indicate the large bar. It would make the rest of the figure more readable.

Figure 11: In the figure, it is not clear what is seawater – partly because it is zoomed in so much, and partly because of the lack of color. Please fix the figure to clarify this. Consider including a "because. . ." at the end of the caption to explain the conservative estimation.

Supplemental data: Include list of Landsat scenes as txt or csv (in addition to PDF) to allow it to be more easily used in the future.

---

## Referee Comment (RC2) · Anonymous Referee #2 · 11 Apr 2016

Overall Assessment

Burton-Johnson et al. present a novel methodology, using freely-available remote sensing data, to perform a classification for the surface of Antarctica. The techniques used in the manuscript improve on existing methodologies which have inconsistencies for the presented problem (automated differentiation between rock, ice/snow, and water). I have included further comments / questions below, but ultimately recommend this manuscript for publication with minor revisions addressed.

Impact

I'd like to comment about the availability and quality of existing Antarctic geospatial (vector) datasets; the Antarctic Digital Database (ADD) has been the de facto standard for open, continent-wide, generally non-scientific base data layers (e.g. coastlines, lakes, rock outcrops, contours, etc.). With this manuscript and the resulting vector dataset of rock outcrops, the authors have contributed significantly to the improvement of Antarctic mapping and geospatial data. Moreover, the methodology presented here allows for the continued refinement of the aforementioned dataset using more recently acquired Landsat 8 imagery or imagery from other, higher-resolution multispectral optical sensors. Although some parameters may need to be revisited for other sensors, the authors presentation of the methodology and delivery of the ArcPy script provide a great launch point for further application (even for novice remote sensing scientists).

Specific Comments/Questions

1) Data Selection The Landsat 8 OLI sensor is an appropriate sensor for this study, given its spatial resolution, revisit frequency, multispectral bands, and cost. Notwithstanding the data incapacity at extreme southern latitudes, can the authors comment on the selection of individual images to be used? Did the authors set any threshold for to certain sun elevations, time of year, or cloud-cover percentage [mentioned P1 L25] Also, given that Antarctic's rock can be covered in snow at any time, were there efforts to exclude those types of images by manual inspection? If either case, for the selection of images in the study, these thresholds should be noted.

2) Accuracy Assessment I believe that the authors have provided a thorough assessment of the accuracy of their methodology and succinctly describe its use and limitations. Can the authors comment if any ground-based verification has been completed? For example, spectrometer samples from the various classification types (e.g. shaded rock, shaded ice) would verify the spectral signatures and further refine (or confirm) the threshold values used.

[Figure]

3) Total Outcrop Area The total outcrop area, I anticipate, is going to be highly cited. Please provide the methodology or source for calculating the "total land area of Antarctica." The final result of 0.18% will vary based on that value. Moreover, it may be beneficial to provide error bars for the final figure.

4) Methodology The authors are lacking sufficient explanation of the dataset merging procedure, especially for overlapping tiles. The authors state that for overlapping tiles if any of the "pixel stack" was classified as rock it was included as rock. Please provide justification for this technique. Furthermore, I believe that the algorithm could be greatly improved with the inclusion of many overlapping tiles. This would remove outliers (e.g. seasonal snow differences) and offer a measure of statistical significance; for example for 5 overlapping images all 5 images provided the same result, that pixel would be assumed to reduce both omission and commission disagreements.

5) Future Considerations Please note that in Section 4.3, many of these satellites have already launched, not "under development or planned for launch" – please update this for the currency of publication date. I do appreciate the authors' consideration for higher resolution datasets and that this technique is not sensor specific (although does have certain requirements, e.g. band availability).

Technical Comments [P2 L5] "several" seems unnecessary [P2 L15] "more strongly" –> "stronger" [P2 L33] Remove extra space after "ablation" [P3 L26] How do you define "strong illumination" and "minimal cloud cover" [P5 L26] "LANDSAT" –> "Landsat 8" [P8 L13] "Digitalglobe's" –> "DigitalGlobe" [P8 L13] "Worldview-3" –> "WorldView-3" [P8 L26] Period, not comma, before "The new map, . . ." [P9 L1] Note the acknowledgements section is included twice in the manuscript [Fig 4] The box containing "create a new raster for sunlit rock..." should read "four" requirements, not "three" [Fig 7] These figures, in general, are very difficult to understand given the scale of the map and density/overlap of the outlines. Although the authors' intention is valid, the detail provided by the outlines are indiscernible for several of the figures. Moreover, the underlying satellite imagery is often covered by the outlines. I suggest reducing the number of

examples and subsequently enlarging them to provide the reader with more detail to better communicate the purpose of the figure. [Fig 8] Should the new rock outcrop dataset only include areas where there is tile coverage? Can the authors be certain that there are no outcrops >82°40'S (the stated domain) that do not have a tile for this analysis? For example, there is a tile gap on the margin (Bryan Coast, Ellsworth Land). [Fig 11] It is very reasonable for manual digitization to clean up the dataset. Can the authors provide the areas that were manually edited after the analysis? If that metadata is unknown, the reproducibility for a given tile is in question.

---

## Author Comment (AC1) · 27 May 2016

Dear Allen Pope,

Many thanks for your helpful and thorough review of our submitted manuscript to The Cryosphere. We have addressed each of your comments below and in the revised manuscript (included along with the additional revised supplementary material in the supplement to this reply).

[Figure]

Gratefully,

Alex Burton-Johnson

Authors' response to Review 1:

General Comments

Burton-Johnson et al. present a new, largely automated method for identifying rock outcrops across all of Antarctica using Landsat 8 data. This paper contributes both a well-documented, open methodology as well as the resulting dataset. I wanted to especially commend the authors on their commitment to reproducible and open science. The paper is clearly, concise, and open and will surely be widely used by the community (I know I will cite it when I update "Open Access Data in Polar and Cryospheric Remote Sensing", Remote Sensing, 2014). I look forward to seeing it fully published in The Cryosphere. Nevertheless, some edits could be made to make this method more applicable to future studies, to slightly expand the region over which Landsat data are applied, and to stylistically clarify a few points. See below for more specific comments.

- Many thanks for these positive comments. We hope to see the open access data for Polar research continue to increase and we hope that this contribution helps forward that goal.

Specific Comments

1) You include a short discussion of "how robust to a new sensor," but there are specific points you can include in the paper to better ensure this. Specifically, use units rather than DNs for thresholds (K, % reflectance), and also describe specifically how each threshold was set. For example, it appears that the NDSI threshold was the 95th percentile of sunlit rock. Including this would allow this method to be applied for future Landsats (to look for change), or with other sensors (to not rely on the ADD where Landsat coverage is unavailable).

- The TOA and brightness corrected Landsat 8 products provided by espa.cr.usgs.gov

are scaled for storage in a 16-bit raster. This is discussed now in the "Methodology" section and both the scaled and non-scaled values are now provided in the text to aid readability of the document and application of the methodology by the reader. Non-scaled values are now included in the thresholds figure, Fig. 5. The derivation and definition of each threshold value is now described in the Methodology section.

2) The authors include a discussion and figure to illustrate the paucity of medial moraines in Antarctica. While this is largely true, I believe some manual cleaning must still be done. For example, in blue ice areas / the Dry Valleys, there may be some medial moraines which should be removed. A quick investigation of the Ferrar Glacier indicates this may be the case there.

The presence of localised occurrences of glacial debris cover mapped as outcrop is now acknowledged in the "Introduction" and "Limitations" sections: "4. Whilst Antarctic glaciers are rarely show any debris cover (Fig. 4) there are local occurrences where extensive debris cover does occur (most notably in the vicinity of the Dry Valleys and the NW coast of the Ross Ice Shelf) which are mapped as outcrop in the new dataset. However, it should be noted that these occurrences are isolated on the continental scale. As this project aims to provide a consistent and automated approach that can be reproduced in the future (for example to monitor change in ice cover over time or season) our methodology attempts to be as free as possible from manual changes. We accept that in some areas localized occurrences of debris covered glaciers may need to be manually altered if detailed topographic maps of rock outcrop are required."

3) Regarding extent of available Landsat 8 coverage – is there a mechanism in place to allow for updated data collections where Landsat 8 is now available rather than the ADD. Specifically: A very quick EarthExplorer search showed that, for example, LC80021082016077LGN00 could possibly be used for Peter 1 Island. I would expect this to be the case for other sub-Antarctic islands. Please consider checking again for new Landsat 8 acquisitions, and discuss if there will be an updating procedure. There have been very recent Landsat 8 off-nadir acquisitions which extend Landsat coverage

further south (and more are planned for next year). Could you include these in your analysis to reduce the areas where the ADD is needed? Will you be able to update next year to extend, too? Or is that prohibitive?

- It will not be possible to update the dataset as individual images are acquired, but it is intended to be updated in the future once a significant improvement has been made in image coverage, especially for the Antarctic islands and off-nadir regions. The dataset and the procedure will be incorporated into the ADD and consequently future iterations of the ADD will aim to widen the image coverage. This is now discussed at the end of the "Applications and future developments" section: "Once the available imagery has improved the Antarctic rock outcrop dataset will again be updated to exploit the new data and increase coverage of the continent (especially south of 82°40' S). By providing the code used in this study (GitHub, github.com/mblack2xl/AntarcticRockOutcrop) users will be able to apply the new methodology to their specific areas of interest and modify the thresholds for improved results on local scales. Such work may be possible to integrate in to future iterations of the continental dataset if users inform the authors regarding their new datasets."

Technical Corrections

Title: This is not technically a fully automated method (manual cleaning is performed) – please edit title.

- The word "fully" is removed from the title and text. However, we argue that this term does not require further dilution as the methodology presented is automated, the manual intervention only being performed because of inadequacies in the Antarctic coastline dataset rather than the new methodology. Should the coastline be better mapped the methodology would be fully automated, and remains so away from the coast.

Title: "Landsat" should be "Landsat 8"

- Corrected

Abstract L9: I would suggest including an introductory sentence that provides motivation for the study.

- The following sentence is included: "As the accuracy and sensitivity of remote sensing satellites improve there is an increasing demand for more accurate and updated base data sets for surveying and monitoring."

Abstract L14: This is not a "fully automated" method – there is some manual cleaning. Please edit, perhaps to "largely" "nearly" or "overwhelmingly"

- "almost" included

Abstract L18: May need to edit latitude (see Specific Comments)

- Reworded to "merged with existing data where Landsat 8 tiles are unavailable; most extensively south of 82°40' S". This is discussed further under Specific Comments.

Abstract L19: please indicate what physical areas 0.18% corresponds to.

- Extent of outcrop included in the text and abstract (21,745 km2)

P1 L21: I suggest citing appropriate examples where the ADD has been used.

- Three examples given of publications on different subjects (glaciology, geological mapping and geophysics) using the ADD rock outcrop data (Golynsky et al., 2006; Riley et al., 2011; Vaughan et al., 1999)

P1 L26: Should "data" be treated as singular or plural?

- This is a classic point of discussion and controversy based on whether "data" represents "facts" (plural) or "information" (singular). Copernicus and the EGU journals do not specify whether they view "data" as a singular or plural noun. We have taken the singular here based simply on its readability and our personal preference, but understand that other readers may feel differently.

[Figure]

P1 L27-28: Parenthetical note is perhaps presented too early for smooth flow of an introduction.

- Parenthetical section removed

P1 L21-29: You should include some durable citation for the ADD if at all available. And/or a citation describing the dataset.

- Citation added (Thomson and Cooper, 1993)

P2 L1: Consider adding a figure to illustrate the issues in the ADD dataset?

- Figure added to illustrate the issues of georeferencing, generalisation and overestimation in the existing ADD rock outcrop dataset.

P2 L1: Comma after "Additionally"

- Comma added

P3 L8: "Landsat" should be "Landsat 8"

- Changed to "Landsat 8"

P3 L19: Follow parallelism in sentence structure – ". . .and be divided into sufficiently large scenes to allow for manual selection of suitable tiles for the entire continent."

- Replaced with suggested text.

P4 L1: Note that when building the Landsat Image Mosaic of Antarctica, Bindschadler et al. used TOA reflectance as well, given then largely dry and thin atmosphere over Antarctica. This could be a good supporting reference for TOA, if you wanted to include one.

- It doesn't hurt to include a supporting reference for your decisions! The LIMA reference is now included: ". . .rendering surface reflectance corrected data unsuitable. Instead, top of atmosphere reflectance and brightness temperature corrected products were used, as were also used for the Landsat Image Mosaic of Antarctica (Bindschadler et al., 2008).

P4 L10: I love that you're sharing your code! Perhaps also consider putting it on GitHub to allow sharing, editing, forking, etc.?

- Hosted on GitHub with a URL included in the paper (github.com/mblack2xl/AntarcticRockOutcrop)

P5 Eqn2: You use the "TIRS" acronym, so to be consistent I think you should also use the "OLI" acronym, rather than referring to it as the Landsat 8 platform.

- "OLI" included for consistency both on these lines and elsewhere in the document with definitions of "OLI" and "TIRS" included in the section "Input Data".

P5 L8: Please clarify whether "liquid water" refers to ocean, on land, or melt ponds on ice (or all of the above). It is a little confusing right now.

- The following definitions included in this paragraph: "...procedures to exclude liquid water offshore (seawater) and onshore (melt ponds)" and "...was also used as a mask for excluding seawater and sea ice".

P5 L11-12: What is "blue intensity"? Reflectance? Some index?

- Changed to "By comparing the blue reflectance values of pixels representing rock and snow..."

P5 L23-24: Aren't all Antarctic satellite images already provided in Antarctic Polar Stereographic? Why is any reprojection necessaru?

- Not for scenes with a centre latitude greater than or equal to -63° S (e.g. the South Shetland Islands), as is now stated in the text: "Tiles not already projected with the WGS 1984 Stereographic South Pole spatial reference (i.e. those at scenes with a centre latitude greater than or equal to -63° S) were then reprojected to this projection before the results of all the tiles were mosaicked together for the entire continent."

P5 L26: "Landsat", not "LANDSAT"

- Corrected

P6 L3-6: Include total area used for this? What about an error matrix for this test?

- Total area and pixel count included. Given that the QC used 10 different images and determined mean values from them, an error matrix would not be suitable. However, we have now included a summary table of the mean QC statistics.

P6 L11: reference formatting

- Reference corrected

P7 L14: See Specific Comments.

- See reply in "Specific Comments"

P7 L15: Have you considered including some quality information in the dataset, which indicates the date/image from which a particular polygon was digitized?

- This was not included here as most outcrop polygons were derived from multiple overlapping polygons. However, should/when this dataset is updated such attributes would be invaluable to trace the development of the dataset.

P7 L20: Somewhere in the discussion, it would be good to acknowledge the issues that TIRS has been having, its recalibration efforts, etc. This is important if your TIRS thresholds are at all sensitive. . .

- The calibration issues are now acknowledged in the Input Data section: "However, the TIRS has suffered from calibration issues and whilst calibration changes have been made to some of the TIRS1 datasets, the TIRS2 data has a larger and more variable calibration uncertainty. Consequently only TIRS1 data is used in this study."

P7 L27: What was used as "truth" for this manual removal?

- This distinction could only be made by eye, as is now discussed in the manuscript.

P8 L7: Include space after "km2" and before "total"

- Space added

P8 L10-17: See Specific Comments. The modification and selection of thresholds should be discussed more in depth in methods to facilitate this cross-application.

- See reply in "Specific Comments"

P8 L26: Period after "respectively" (not a comma)

- Corrected

P8 L27: May need to amend sentence with information about subantartic islands / off-nadir acquisitions (see Specific Comments).

- As noted in Specific Comments, we acknowledge that there are sparse off-nadir images slowly being acquired. Hopefully this will facilitate an update of this dataset in the future, but we do not intend to update it piecemeal as each image is acquired.

P8 L27: Include are which 0.18% corresponds to.

- Area added

P8 L28: ". . .48% of the previous estimate (0.37%, _______km2)."

- Area and percentage added

Figure 1: The snow reflectance line is solid in the legend but dashed in the figure. Make it solid in the figure. Consider using line colors from ColorBrewer to help colorblind readers.

– The snow reflectance plot is now a solid line. The colours remain unchanged as for the wavelength boxes they relate to the RGB wavelength they represent (blue for blue, etc. . .) and having looked at ColorBrewer, the different warm shades (red and yellow) for the spectra should not be a problem for colour blind readers. Many thanks for drawing this facility to our attention though – it will be very useful for future publications

and maps.

Figure 2: Where did these data come from? Are the shadowed areas rock, it is hard to the viewer to see.

- False colour image (in the combination Red: SWIR2; Green: Blue; Blue: Blue) now used to help the reader differentiate rock, snow and clouds. The scene ID is now in the figure caption.

Figure 3: I don't find this figure very helpful – especially given Specific Comment regarding Ferrar Glacier. Could be removed or combined with other figure I suggest (illustrating issues with ADD).

- There are localised areas of debris cover on Antarctic glaciers, of which the Dry Valleys region is the most notable, likely associated with the presence of extensive outcrop and liquid water in the region. However these are exceptional occurrences on the continental scale, and it is important that readers unfamiliar with this feature (especially those who work at lower latitudes) have this property illustrated to them. From experience, those who work with lower latitude glaciers view debris cover as their principal issue when detecting glacial extent making them sceptical of work at higher latitudes where this is not such a problem, and so we recommend keeping this figure in the final manuscript. The issue of debris cover is now acknowledged in the "Limitations" section (see "Specific Comments").

Pre-calculated Figure 4: In general, I don't see a systematic use of boxes and arrows to indicate particular inputs/outputs/actions. For example, rather than have an arrow pointing at "Identify Sunlit Rock" or "Identify Shaded Rock," these should be larger boxes (possibly shaded background) which encompass the specific tasks which achieve those goals.

- As is the standard for flowchart design, the parallelogram is used initially to represent the input data, the oblongs are used to indicate the start and end of the program
flow (for both sunlit and shaded rock outcrops) and the intermediate processes are all indicated in rectangular boxes. The "Identify Sunlit Rock" and "Identify Shaded Rock" boxes are now shaded to make them stand out and more visually separate the two processes.

In the top box "corrected" is not the right word, I don't think. "Pre-calculated" or just removed it, perhaps?

- Changed to "brightness temperature converted" and left included in the figure as is one of the options users must select when requesting the tiles for download.

Relatedly, when are mask arrows being added or removed from each other? Clarify.

- Addition of masks now specified on flowchart.

Units should be used either instead of or in addition to the DN values for TIRS and Blue.

- Scaling removed from units and units now defined on the y-axes.

Remove superfluous steps (e.g. possibly reprojection, or "repeat steps for all Landsat images" What method specifically was used for mosaicking Landsat rasters?

- Both "reprojection" and "repeat. . ." are required steps, so remain in the workflow.

What process used for overlapping areas?

- A paragraph is now included in the "Methodology" section describing the raster mosaicking process.

What connectivity or filtering was used in polygonization?

- As the raster is composed of 30 m wide, square-sided pixels no filtering was required. Instead, the extent of the binary raster mosaic could be converted in to a new polygon using the ArcGIS "raster to polygon (conversion)" tool. Some minor modifications are made to the text.

Rather than clipping ADD to a particular latitude, consider clipping to specifically areas of no Landsat?

- The ADD was clipped using the footprints of the Landsat 8 tiles. The Latitude of their southerly extent was used to describe this in the text but as this is less specific the text and figure have now been corrected.

Caption can deleted "using the new methodology" – this is implicit.

- Text removed from caption.

Figure 5: Regarding y axis units – some are clear (e.g. NDSI, NDWI), but it would be good to clarify the units/scaling factors for TIRS, Blue, etc. Also, again, what is "Blue Intensity"? Do you mean reflectance? Finally – how were test areas selected? And are "n" values pixels?

- The TIRS1 brightness temperature, blue reflectance and TIRS1/Blue values have been corrected for the scaling factor and labelled as such. Scaling is discussed in Section 2.2 and referenced as such in the caption. "n" is now defined in the caption. Test area selection is now defined in the body text.

Figure 6: See Specific Comment regarding islands and coverage.

- See reply in Specific Comments. The caption is also corrected to state that the ADD outcrop dataset was used for areas without Landsat 8 coverage rather than a specific latitude.

Figure 7: Include NDSI threshold either annotating figure (or in supplementary information if you think that would be too crowded). Also, consider organizing a-j by type of scene (or labeling as such) to make translation between text (P6 L11-16) and the structure of the figure easier. Two specific comments: "tree" should be "three" and "Automated Outcrop Identification" should be "Automated Outcrop Identification (presented here)"

- The NDSI threshold is now shown in brackets on each figure. Reviewer 2 requested that the figure size be increased, so this change has not resulted in overly crowded images. They figures have also been reorganised as suggested to match the discussion in the text and make the figure easier to follow. The word "tree" has been corrected to "three" and the Legend has been modified to include "Automated Outcrop Identification (presented here)".

Figure 8: How were areas chosen to not use any Landsat imagery? What if outcrops are showing up in new locations? Is there a method to handle that sort of change? Or inform selection of imagery from other sensors?

- Whilst not as accurate as the new rock outcrop dataset, the existing ADD dataset was produced to highlight all areas of outcrop, even exaggerating the size of very small areas so that they could be seen and liberally classifying areas of shaded snow in case they contained rock outcrop. Consequently we can be confident that all outcrops on the continent detectable by the new method are represented in the old dataset. This meant that we could use the existing ADD outcrop dataset to select the Landsat 8 tiles based on the Landsat tiles' classification grid, the World Referencing System (WRS) and analyse just these tiles rather than having to analyse tiles over the entire continent (a process that may have made this work far more time consuming, potentially unfeasible, and increase the opportunity for commission misclassification).

Figure 9: Should "Jr" have a period after it in the citation?

- Period added

Figure 10: I would cut off the y axis of part b at 2 or a little below and include a label to indicate the large bar. It would make the rest of the figure more readable.

- Y axis cut off at 2 with the cropped error bar now labelled to indicate its maximum value.

Figure 11: In the figure, it is not clear what is seawater – partly because it is zoomed

in so much, and partly because of the lack of color. Please fix the figure to clarify this. Consider including a "because. . ." at the end of the caption to explain the conservative estimation.

- False colour images are now used to accentuate the rock, snow and seawater pixels (in the combination Red: SWIR2; Green: Blue; Blue: Blue). An explanation of the conservative chosen threshold values is also included at the end of the caption.

Supplemental data: Include list of Landsat scenes as txt or csv (in addition to PDF) to allow it to be more easily used in the future.

- List of scenes now also included as a tab delimited text file as well as a PDF.

New References Bindschadler, R., Vornberger, P., Fleming, A., Fox, A., Mullins, J., Binnie, D., Paulsen, S. J., Granneman, B. and Gorodetzky, D.: The Landsat image mosaic of Antarctica, Remote Sens. Environ., 112(12), 4214–4226, 2008. Golynsky, A., Chiappini, M., Damaske, D., Ferraccioli, F., Finn, C. A., Ishihara, T., Kim, H. R., Kovacs, L., Masolov, V. N., Morris, P. and others: ADMAP—a digital magnetic anomaly map of the Antarctic, in Antarctica, pp. 109–116, Springer. [online] Available from: http://link.springer.com/chapter/10.1007/3-540-32934-X_12 (Accessed 11 April 2016), 2006. Riley, T. R., Flowerdew, M. J. and Haselwimmer, C. E.: Geological Map of Eastern Graham Land, Antarctic Peninsula (1:625 000 scale), British Antarctic Survey, Cambridge, UK., 2011. Thomson, J. W. and Cooper, A. P. R.: The SCAR Antarctic digital topographic database, Antarct. Sci., 5(03), 239–244, 1993. Vaughan, D. G., Bamber, J. L., Giovinetto, M., Russell, J. and Cooper, A. P. R.: Reassessment of net surface mass balance in Antarctica, J. Clim., 12(4), 933–946, 1999.

Please also note the supplement to this comment:
http://www.the-cryosphere-discuss.net/tc-2016-56/tc-2016-56-AC1-supplement.zip

---

## Author Comment (AC2) · 27 May 2016

Dear Reviewer #2 (Anonymous),

Many thanks for your helpful and thorough review of our submitted manuscript to The Cryosphere. We have addressed each of your comments below and in the revised manuscript (included along with the additional revised supplementary material in the supplement to this reply).

[Figure]

Gratefully,

Alex Burton-Johnson

Authors' response to Review 2:

Overall Assessment

Burton-Johnson et al. present a novel methodology, using freely-available remote sensing data, to perform a classification for the surface of Antarctica. The techniques used in the manuscript improve on existing methodologies which have inconsistencies for the presented problem (automated differentiation between rock, ice/snow, and water). I have included further comments / questions below, but ultimately recommend this manuscript for publication with minor revisions addressed.

- Again, many thanks for your positive comments and review.

Impact

I'd like to comment about the availability and quality of existing Antarctic geospatial (vector) datasets; the Antarctic Digital Database (ADD) has been the de facto standard for open, continent-wide, generally non-scientific base data layers (e.g. coastlines, lakes, rock outcrops, contours, etc.). With this manuscript and the resulting vector dataset of rock outcrops, the authors have contributed significantly to the improvement of Antarctic mapping and geospatial data. Moreover, the methodology presented here allows for the continued refinement of the aforementioned dataset using more recently acquired Landsat 8 imagery or imagery from other, higher-resolution multispectral optical sensors. Although some parameters may need to be revisited for other sensors, the authors presentation of the methodology and delivery of the ArcPy script provide a great launch point for further application (even for novice remote sensing scientists).

- As with our reply to Reviewer 1, we hope that providing the new dataset and ArcPy script will help improve increasing availability of open access data for Polar research.

Specific Comments/Questions

Data Selection

The Landsat 8 OLI sensor is an appropriate sensor for this study, given its spatial resolution, revisit frequency, multispectral bands, and cost. Notwithstanding the data incapacity at extreme southern latitudes, can the authors comment on the selection of individual images to be used? Did the authors set any threshold for to certain sun elevations, time of year, or cloud-cover percentage [mentioned P1 L25] Also, given that Antarctic's rock can be covered in snow at any time, were there efforts to exclude those types of images by manual inspection? If either case, for the selection of images in the study, these thresholds should be noted.

- Where possible, images were selected that were taken during the Austral summer between September and March; had solar elevation angles >20°; and have <30% cloud cover, with only a small number of tiles having to be used that do not meet these requirements in areas where better data did not exist at the time. This is now detailed in "Section 2.1 Input Data". We note that the amount of snow cover on Antarctica's rock outcrops can be variable, and this is resolved to some degree by the use of multiple images being analysed for most areas of the continent (also detailed in "Input Data" and the section: "Procedure C. Applying the masks and merging the datasets").

Accuracy Assessment

I believe that the authors have provided a thorough assessment of the accuracy of their methodology and succinctly describe its use and limitations. Can the authors comment if any ground-based verification has been completed? For example, spectrometer samples from the various classification types (e.g. shaded rock, shaded ice) would verify the spectral signatures and further refine (or confirm) the threshold values used.

- Ground-based spectral data does exist, and at the coarse resolution of Landsat imagery, spectral library measurements would also be valid. However, this project aimed
to develop an in-scene technique using thresholds defined on TOA reflectance data.

Total Outcrop Area

The total outcrop area, I anticipate, is going to be highly cited. Please provide the methodology or source for calculating the "total land area of Antarctica." The final result of 0.18% will vary based on that value. Moreover, it may be beneficial to provide error bars for the final figure.

- The classification accuracy is now included, taking in to account the omission and commission errors. This now allows determination of the error on the total outcrop area figure (0.18% $\pm$0.05%). Modifications are included in the text and in Fig. 10 to account for this, including the equation defining the accuracy calculation and the inclusion of the mean classification accuracy values in Fig. 10a.

Methodology

The authors are lacking sufficient explanation of the dataset merging procedure, especially for overlapping tiles. The authors state that for overlapping tiles if any of the "pixel stack" was classified as rock it was included as rock. Please provide justification for this technique. Furthermore, I believe that the algorithm could be greatly improved with the inclusion of many overlapping tiles. This would remove outliers (e.g. seasonal snow differences) and offer a measure of statistical significance; for example for 5 overlapping images all 5 images provided the same result, that pixel would be assumed to reduce both omission and commission disagreements.

- Details of the mosaicking process and justification for the use of overlapping tiles are now included in the text: "As most areas were covered by multiple overlapping Landsat tiles, any pixels identified as rock exposure by any of the overlying tiles was included as exposed rock in the final dataset. This was achieved by mosaicking the binary raster files produced by the workflow and taking the maximum pixel value. If a pixel was classified as snow it was designated "0" by the script, or "1" if it represents
rock. Consequently this mosaicking process stores rock outcrop pixels ("1") in the raster mosaic in preference to snow ("0"). By analysing multiple overlapping tiles the methodology becomes more sensitive to identifying rock outcrops; allows detection of rock outcrops even when they are obscured by clouds in one tile of the input data; and makes the methodology less sensitive to seasonal or short term variation in snow cover." We also like the suggestion of using the number of overlapping tiles as a way of assigning a degree of statistical significance and will consider it in future versions of the dataset. One issue is that a number of areas are only covered by very few tiles due to cloud cover, especially on the Antarctic Peninsula, so a script using this methodology would need to assign a degree of significance to each individual pixel and details of this stored in its metadata. This should be feasible and is definitely something to consider.

Future Considerations

Please note that in Section 4.3, many of these satellites have already launched, not "under development or planned for launch" – please update this for the currency of publication date. I do appreciate the authors' consideration for higher resolution datasets and that this technique is not sensor specific (although does have certain requirements, e.g. band availability).

- Details of each satellite program's progress have been included and updated.

Technical Comments

[P2 L5] "several" seems unnecessary

- Word deleted

[P2 L15] "more strongly"–> "stronger"

- Reworded as recommended

[P2 L33] Remove extra space after "ablation"

- Space deleted

[P3 L26] How do you define "strong illumination" and "minimal cloud cover"

- The following text is now included: "To ensure strong illumination we only used images taken during the day in the Austral summer between September and March. An estimate of cloud cover is included in the metadata for Landsat 8 images, and for all but four scenes images could be used with less than 30% cloud cover."

[P5 L26] "LANDSAT" –> "Landsat 8"

- Corrected

[P8 L13] "Digitalglobe's" –> "DigitalGlobe"

- Corrected

[P8 L13] "Worldview-3" –> "WorldView-3"

- Corrected

[P8 L26] Period, not comma, before "The new map, . . ."

- Corrected

[P9 L1] Note the acknowledgements section is included twice in the manuscript

- Repeated section removed.

[Fig 4] The box containing "create a new raster for sunlit rock..." should read "four" requirements, not "three"

- Wording corrected

[Fig 7] These figures, in general, are very difficult to understand given the scale of the map and density/overlap of the outlines. Although the authors' intention is valid, the detail provided by the outlines are indiscernible for several of the figures. Moreover, the underlying satellite imagery is often covered by the outlines. I suggest reducing the number of examples and subsequently enlarging them to provide the reader with more

detail to better communicate the purpose of the figure.

- Each figure is discussed in section 3.1 so should ideally be shown in the paper. Consequently we have resized the figure to take two pages whilst still keeping the total article length concise. The images are enlarged further (one image per page) in the PDF document now included as supplementary material.

[Fig 8] Should the new rock outcrop dataset only include areas where there is tile coverage? Can the authors be certain that there are no outcrops >82âŮę40'S (the stated domain) that do not have a tile for this analysis? For example, there is a tile gap on the margin (Bryan Coast, Ellsworth Land).

- The new outcrop dataset could have been left with just the areas with tile coverage and not completed for the continent using the existing ADD rock outcrop dataset. However, most users would require a continent-wide dataset and so it was decided that splicing the two datasets would greatly increase the usability of the new work. As off-nadir images start to be collected we hope that an update of this dataset in the future will reduce our reliance on the old data. Although the existing ADD rock outcrop dataset is less accurate than the new, Landsat 8 derived data, it is very good at highlighting any areas of outcrop on the continent, even exaggerating the areas of small outcrop. Consequently we were able to select tiles covering all exposures of rock on the continent and by visually comparing the tile areas with the LIMA mosaic are confident that this is the case.

[Fig 11] It is very reasonable for manual digitization to clean up the dataset. Can the authors provide the areas that were manually edited after the analysis? If that metadata is unknown, the reproducibility for a given tile is in question.

- Some misidentified pixels were present in all coastal scenes containing seawater. This is now specified in the text. However, as no manual editing was done on land the repeatability of this methodology should not be affected (this is now included in the text).
Please also note the supplement to this comment:
http://www.the-cryosphere-discuss.net/tc-2016-56/tc-2016-56-AC2-supplement.zip
* * *